# From the Cytoplasm into the Nucleus—Hepatitis B Virus Travel and Genome Repair

**DOI:** 10.3390/microorganisms13010157

**Published:** 2025-01-14

**Authors:** Johan Ringlander, Gustaf E. Rydell, Michael Kann

**Affiliations:** 1Department of Infectious Diseases, Institute of Biomedicine, Sahlgrenska Academy, University of Gothenburg, 41346 Gothenburg, Sweden; johan.ringlander@gu.se (J.R.); gustaf.rydell@gu.se (G.E.R.); 2Department of Clinical Microbiology, Region Västra Götaland, Sahlgrenska University Hospital, 41346 Gothenburg, Sweden

**Keywords:** hepatitis B virus, core protein phosphorylation, capsid, intracellular transport, nuclear import, nuclear pore complex (NPC), genome release, genome repair

## Abstract

Hepatitis B virus (HBV) is a major global health concern, affecting millions of people worldwide. HBV is part of the hepadnaviridae family and one of the primary causes of acute and chronic liver infections, leading to conditions such as cirrhosis and hepatocellular carcinoma (HCC). Understanding the intracellular transport and genome repair mechanisms of HBV is crucial for developing new drugs, which—in combination with immune modulators—may contribute to potential cures. This review will explore the current knowledge of HBV intracytoplasmic and nuclear transport, as well as genome repair processes, while drawing comparisons to other viruses with nuclear replication.

## 1. Preface

The study of HBV replication faces significant challenges due to the limited availability of suitable animal models. HBV exhibits strict host and tissue specificity, with chimpanzees being the only animals naturally susceptible to human HBV infection. While animal hepatitis viruses with similar replication strategies exist, they also exhibit key differences. For instance, studies on duck hepatitis B virus (DHBV) have advanced our understanding of the HBV replication cycle, particularly the initiation of liver infection. However, DHBV infection is restricted to ducklings under three weeks old (typically one day old), and hepatocellular carcinoma (HCC) does not develop without additional carcinogenic factors [1]. Similarly, woodchuck hepatitis virus (WHV) shares many aspects of HBV replication. However, chronic inflammation seems to be less important, and specific viral genome integration events in host chromatin leading to c-myc expression are more important for HCC development than in HBV-infected humans [1].

To address these limitations, researchers have developed immune-deficient mouse models with partially humanized livers [2]. These models rely on the partial repopulation of the mouse liver with primary human hepatocytes. While these systems enable HBV infection studies, they have significant drawbacks, including technical complexity, inconsistent infection outcomes depending on hepatocyte quality, and an inability to replicate immune responses or study physiological HCC development.

Most hepatocyte-derived cell lines are naturally resistant to HBV infection, requiring transfection with HBV DNA for research purposes. Notable exceptions are HepaRG cells and cell lines genetically modified to express the HBV-specific receptor NTCP (sodium taurocholate co-transporting polypeptide). However, these lines require high multiplicities of infection (MOIs) and often produce only limited amounts of virus, typically not exceeding the input levels used during infection. This is in striking contrast to in vivo infections, where ten or fewer HBV virions are sufficient to cause infections, as shown in chimpanzees [3].

These constraints have necessitated the use of alternative experimental systems to investigate specific aspects of the HBV life cycle. For example, microinjection into *Xenopus laevis* oocytes has been employed to study intracellular transport, while digitonin-permeabilized cells are used to analyze nuclear import processes. Additionally, to investigate the structural and functional properties of HBV capsids—which require high purity and concentration—researchers often rely on capsids produced via expression systems in *E. coli* or mammalian cell cultures.

### Overview of the HBV Life Cycle

The infectious hepatitis B virus (HBV) particle contains a partially double-stranded DNA genome approximately 3.2 kilobases long, packaged in a relaxed circular form (rcDNA) within a (nucleo) capsid. Within the virion, the capsid is enveloped by a layer formed by three surface proteins, termed large (L), middle (M), and small (S) HBs, which are embedded into lipids and share a common C-terminus. Recent observations of Liu et al. showed that these lipids were not organized in a bilayer as previously thought but in a noncanonical lipid patch, which can accommodate the exposed hydrophobic surface and modulate particle stability [4].

HBV employs a distinctive replication mechanism that involves reverse transcription of pregenomic RNA (PG) and nuclear transport, both critical for its life cycle and persistence in hepatocytes [5]. Additionally, a minority (~10%) of HBV virions harbor double-stranded linear DNA (dslDNA) [6,7], typically linearized within the direct repeat 1 (DR1). These dslDNA-containing virions undergo similar entry and transport processes as rcDNA virions, but dslDNA becomes efficiently integrated into host chromatin. However, the linearization prevents transcription of PG as it separates the promoter from the gene [8]. This abolishes expression of the capsid protein (core protein; Cp) and the viral polymerase (pol), rendering them incapable of producing new virions but still allowing expression of the viral surface proteins [8,9]. Furthermore, the reverse transcription of PG occasionally stops; therefore, virus particles containing PG and DNA intermediates with RNA of different lengths are found in serum [10]. In addition, patients’ sera also contain virus-like particles, which are composed of surface proteins and empty capsids. These particles significantly outnumber DNA-containing virions by ~100-fold [11].

HBV enters hepatocytes by clathrin-dependent endocytosis after interactions with heparan sulfate and the HBV-specific receptor NTCP, as shown for NTCP-expressing HepG2 cells [12,13]. Membrane penetration seems to occur in the late endosome either through fusion or membrane permeabilization [14,15,16]. The capsid is released into the cytoplasm (Figure 1 (A)), followed by the transport of the viral genome through the cytoplasm (Figure 1 (B)) and across the nuclear envelope (NE) (for a review, see [5]). Hepatocytes have a long lifespan of at least 200–400 days [17,18], and entry of the rcDNA into the nucleus can thus not occur by passive inclusion into the nucleus during mitosis. Instead, HBV, like most other viruses with a nuclear phase, had to develop a strategy of passing the intact NE (Figure 1 (C)).

Once inside the nucleus, the rcDNA but also a fraction of dslDNA molecules is repaired into a covalently closed circular DNA (cccDNA), forming a minichromosome associated with histones (Figure 1 (D)) [19]. This cccDNA serves as a template for transcription (Figure 1 (E)), which is regulated by several viral and non-viral proteins. The structural maintenance of chromosomes (SMC) complex, particularly SMC5/6, binds to cccDNA, suppressing viral transcription [20]. The viral non-structural X protein (HBx) overcomes this suppression by forming a complex with CUL4-DDB1, which degrades SMC5/6. HBx also recruits histone acetyltransferases (e.g., P300) to regulate transcription through histone acetylation and deacetylation [21]. Although the details are not fully understood, the core protein (Cp) may also interact with cccDNA, influencing transcription [22], and other studies have observed a Cp-independent transcription [23,24].

Although at least 20 spliced forms of the PG and the shorter surface protein mRNA (preS2/S mRNA) are identified [25], HBV replication relies on five unspliced, capped mRNAs, all sharing a common 3′ end. The unspliced mRNAs include three mRNAs shorter than genome length, which encode the three surface proteins and HBx, and two mRNAs of super-genomic length. The longest mRNA encodes the non-structural HBe protein. The second longest, the PG RNA, lacks the HBe start codon and also serves as a template for reverse transcription, and it encodes both Cp and pol. Both proteins are over-expressed with regard to the quantities needed for virion formation.

Pol becomes activated by a complex interplay with heat shock proteins, Cp, and ε (epsilon signal), which is a folded RNA structure on the PG. Heat shock proteins allow ε-binding, preferentially to that of the PG molecule from which pol was synthesized (Figure 1 (F) and Figure 2A) [26]. ε-binding then induces structural changes on pol [27,28], leading to the exposure of its N-terminus (terminal protein [TP]), and allows interaction with the 21.5 kDa Cp—or Cp-dimer—which must be phosphorylated at serine residues 164 and/or 172 [29] (Figure 3). While capsid formation continues via Cp hexamer intermediates to intact capsids [30], the terminal protein mediates the priming of reverse transcription, which is the ligation of the first nucleotide to Tyr63/65 in TP (Figure 2A) (for a review, see [31,32]). Further activity of pol depends upon Cp interaction [33], requiring Cp phosphorylation [33] to an unknown extent, and comprises the synthesis of the first 3–4 nucleotides by copying the bulge within the ε structure and the switch of the pol–nucleotide complex to the DR1 far downstream on PG Figure 2A).

From this site, pol continues with reverse transcription of the PG into ssDNA of minus-strand polarity, which is combined with a degradation of the PG by an RNase H activity on the C-terminal portion of pol (Figure 2B). The minus-strand DNA slightly exceeds genome length by approx. five nucleotides. As the RNase H activity is sterically separated from the catalytic polymerase domain, a small, 13 nt long part, representing the extreme 5′ end of the PG, stays non-degraded. This RNA fragment then translocates to a complementary sequence on the DNA minus strand (direct repeat 2, DR2; Figure 2C), where it serves as the primer for plus-strand DNA synthesis, which then continues until the 5′ end on the minus strand DNA is reached. Due to the terminal redundancy of the minus-strand DNA, the 3′ end of the plus strand can dissociate from the 5′ end of the minus DNA strand for reannealing with the 3′ end of the minus-strand DNA molecule, thereby circularizing the DNA (Figure 2D) and allowing further plus-strand DNA synthesis (Figure 2E). However, the switch of the non-degraded RNA primer does not always occur, and pol starts the plus-strand DNA synthesis by copying the minus-strand DNA to a dsl molecule.

DNA synthesis within the capsids induces structural changes on the capsid surface, which are poorly understood, but the capsid surface must be similar to capsids without any nucleic acid content. This hypothesis can be concluded from the observation that capsid interaction with the viral surface proteins occurs either when DNA synthesis has occurred or with empty capsids, which were described to be a significant fraction of circulating HBV.

Capsid encapsidation occurs in the endoplasmic reticulum (ER) or post-ER compartments (Figure 1 (I)), mediated by interactions between the capsid and a domain at the boundary of preS1 and preS2 of the large surface protein [34,35]. The subsequent virion secretion (Figure 1 (J)) is facilitated by the ESCRT machinery [5] and terminates plus-strand synthesis by restricting deoxynucleotide access [36], yielding the characteristic rcDNA genome.

In contrast with most other viruses, progeny rcDNA-containing capsids are similar or even identical to capsids released from the endocytotic infection pathway. This phenomenon has significant consequences for the HBV life cycle: at the beginning of infection, when the expression of surface proteins is low, capsids may transport the progeny genomes into the nucleus [5], leading to the accumulation of cccDNA to copy numbers between one and nine per infected hepatocyte [37]. Furthermore, infected cells expressing surface proteins are protected from further HBV superinfections, as the capsids derived from new infections become secreted instead of transporting their genome into the nucleus.

The fate of the excessive Cp depends upon the replication activity [38]. In low HBV-replicating cells, capsids are found within the nucleus [5], which are devoid of the viral genome [39]. These intranuclear capsids accumulate and can reach high concentrations in humans but also Cp-transgenic mice [40]. Of note, non-assembled Cp can also be exported back to the cytoplasm, dependent on the Tip-associated factor (TAP), and nuclear export signals (NESs) are found in the Cp C-terminal domain (CTD) [41]. In high HBV-producing cells, the capsids localize in the cytoplasm [42], a phenotype that is also observed in HBV-expressing cell lines [43].

The fate of pol molecules is poorly understood; they are not encapsidated into the capsid. Immune fluorescence studies of the expression of pol alone showed a granular staining in the cytoplasmic space [44], but investigations showed a mitochondrial localization that was independent of Cp [45].

## 2. Core Protein, Capsids, and Their Phosphorylation

### 2.1. Core Protein

As evident from the viral life cycle, the 21.5 kDa Cp with 185 aa (genotype A; 183 aa other genotypes) is crucial. It comprises an N-terminal assembly of around 140 aa and the multifunctional 38 aa long C-terminal domain (CTD; Figure 3). Cp assembles spontaneously to capsids. Assembly is initiated by a rapid Cp dimerization [46,47]. Three Cp dimers form a Cp hexamer, which then assembles into intact capsids [47] with a T = 4 and T = 3 symmetry and diameters of 34 and 32 nm [48], respectively.

The CTD contains four arginine clusters unspecifically interacting with nucleic acids, in particular with RNA [49,50]. In non-phosphorylated capsids, the positive charge of each arginine residue is neutralized by the negative charge of one nucleotide [51].

Between the arginine clusters are six serine residues and one threonine residue, which can be phosphorylated by cellular protein kinases. The nature of the protein kinase is not fully elucidated yet, and several protein kinases are described (Serine-Arginine Protein Kinases 1 and 2 [SRPK-1/-2], Protein Kinase A [PKA], Protein Kinase C [PKC], Casein kinase 2 [CK-2; in particular, CK2-aPKA], Polo-like kinase 1 [PLK-1], Ataxia Telangiectasia Mutated kinase [ATMK], and Cyclin-dependent kinase 2 [CDK-2] [52,53,54,55,56,57,58]. They phosphorylate residues S157, T162, S164, S170, S172, S178, and S180 in the CTD [59] and S44 and S/T49 in the assembly domain [52]. S157, S164, and S172 appear to be the major phosphorylation sites. SRPK-1 caused phosphorylation on all sites, while PKA and PKC phosphorylation was limited to three sites [55]. Phosphorylation of the CTD reduces nucleic acid interaction, resulting in the assembly of empty capsids [55]. In turn, the assembly of non-phosphorylated Cp results in capsids containing unspecific RNA [60]; the presence of the latter enhances assembly [61,62]. As shown by mutation of the serine residues, CTD phosphorylation at 164 and 170 are essential for PG packaging [63].

### 2.2. Capsids

Capsids contain different nucleic acids. Empty capsids—although unspecific nucleic acid packaging cannot always be excluded—are observed in virions, in the cytoplasm, and in the nucleus [64,65]. PG-containing capsids are cytoplasmic; rcDNA- and dslDNA-containing capsids are found in the virion and in the cytoplasm [64], the latter only in marginal amounts upon initial hepatocyte infection.

The Cp in an assembled capsid does not form a close protein shell. It comprises holes of around 1.2 to 2 nm diameter in its 5-fold vertices (12 pores) and quasi 6-fold vertices (30 pores) [66,67] through which nucleotides, which have a Stokes diameter of 5.6 Å, can diffuse into the capsid’s lumen while being in the cytoplasm. Accordingly, envelopment of the capsids by the viral surface proteins is thought to terminate plus-strand DNA synthesis [10,68], which exhibits a variable length in the virion. Furthermore, capsids are immanently unstable. Either Cp dimers or hexamers dissociate from the capsid, followed by their re-association, a process called capsid breathing [69,70,71]. When using capsids expressed in *E. coli*, which are non-phosphorylated, do not contain pol and PG, but have RNA in an identical amount to the PG, a significant proportion of capsids dissociate to Cp dimers and hexamers within 1 h at room temperature. Vice versa, chromatography-purified Cp dimers re-associate to Cp hexamers and capsids under the same conditions [72].

The CTD has an inner and an external topology dependent upon CTD phosphorylation and the nature of the encapsidated nucleic acid. In non-phosphorylated capsids containing RNA, e.g., after expression in *E. coli*, they are found in the capsid’s lumen adjacent to the holes in the capsid wall [73,74]. Introducing negative charges between the arginine clusters by phosphorylation causes repulsion of the CTD, allowing its external localization, and accordingly, empty capsids—even without phosphorylation—also exhibit external CTDs [49].

CTD externalization was shown by the digestion of different capsids with trypsin, which cleaves at arginine residues in the CTD bound to 40 nm gold particles. The phosphorylation of *E. coli*-expressed capsids with PKC and by SRPK1 leads to exposure of parts of the CTD [55,58]. Of note, these experiments showed that the phosphorylated CTDs were exposed, arguing against a structural change, which was derived from the preparation of the capsids. External CTD localization was further confirmed by image reconstruction after cryo-electron microscopy using *E. coli*-expressed capsids [53], a mutant in which serine residues were replaced by acidic glutamate amino acids, mimicking phosphorylation [75]. Aside from confirming the in vitro phosphorylation results, these findings also demonstrate that digestion of the CTD by trypsin was not based on capsid breathing.

This partial CTD exposure was also observed in capsids with rcDNA extracted from HBV from the cell culture supernatant of a cell line stably producing HBV or in non-phosphorylated capsids devoid of nucleic acids; the latter was produced by in vitro dissociation of *E. coli*-expressed capsids, which were re-associated [72]. Interestingly, capsids purified from this cell line after inhibition of HBV genome maturation led to a decreased CTD exposure [61].

Aside from these significant structural changes, the capsids under subtler ones are associated with genome maturation and phosphorylation. Although capsids from human livers are structurally identical to capsids expressed in *E. coli* at a resolution limit of 30 Å and 16 Å [76,77], showing no genome maturation-associated structural differences, a resolution with 10 Å showed that a hydrophobic pocket is present only on DNA-containing capsids [78]. Nonetheless, as these studies were based on cryo-electron microscopy with subsequent averaging, irregular changes such as the exposure of the CTD would have been undetectable.

The extent of phosphorylation is dynamic and varies with the genome content. Cp phosphorylation is high in unassembled Cp and in empty capsids and low in capsids with rcDNA [55]. This implies dephosphorylation of the CTD by a phosphatase, and cellular protein phosphatase 1 (PP1) was identified as a potential candidate [79]. Inhibition and knock-out of PP1 lowered the proportion of hypo-phosphorylated Cp and reduced the amounts of encapsidated PG [79]. Dephosphorylation occurred during capsid assembly. Furthermore, PP1α and β co-sedimented with PG-containing capsids but not empty capsids. As anti-PP1 antibodies failed to precipitate capsids from gradient purification, the authors concluded that PP1 is encapsidated. This interpretation was supported by the detection of PP1α/β in virions from human serum [79].

Others concluded that the pol RNase H domain comprised a phosphatase activity [80]. This hypothesis was based on the observation that capsid dephosphorylation was reduced by pol RNase H mutations and when the ε signal on the PG was mutated. As there was no direct evidence for a phosphatase activity of pol, these data may be in agreement with Cp dephosphorylation by PP1α/β when assuming that the RNase H domain mediates its encapsidation. This study revealed, however, further interesting results for the understanding of Cp phosphorylation dynamics, such as a strong infectivity reduction of virions with heavily phosphorylated capsids, while DNA synthesis and virion secretion are only moderately affected by the capsid phosphorylation status. The latter finding is in agreement with the encapsidation of empty capsids, which are heavily phosphorylated.

It is of interest that the dephosphorylation of empty capsids by phosphatases has also been described to induce capsid disassembly [80].

However, recent findings indicate that the entire picture of capsid phosphorylation and dephosphorylation is even more complex, as capsid re-phosphorylation at both NTD and CTD by packaged CDK2 may facilitate uncoating and covalently closed circular (CCC) DNA formation [79].

Of note, the recent work of Hofmann et al. observed that Cp can also be modified by sumoylation at two lysine residues (K7 and K96), leading to association with PML-nuclear bodies [81]. Infecting NTCP-expressing hepatoma cells with a sumoylation-negative mutant inhibited cccDNA formation significantly, implying that this modification contributes to capsid trafficking and disassembly.

### 2.3. Cytoplasmic Transport

#### 2.3.1. Principles of Active Cytoplasmic Transport

The cytoplasmic movement of larger structures is restricted by viscosity and by numerous organelles and molecular crowding due to high macromolecule concentration around 300–400 mg/mL [82]. Based on determinations using GFP (27 kDa), diffusion is 3–4 times slower than in water, an effect which is even more pronounced for larger molecules such as DNA, which will diffuse 10–100 times slower than in water [83,84]. Furthermore, diffusion is undirected. Larger biological structures, such as viral capsids, thus use active transport processes for their translocation. Eukaryotes provide two transport systems based on cytoskeletal tracks, which are crucial for maintaining cellular organization and function [85]. While actin filaments are generally used for short-distance transport close to the plasma membrane, microtubules are employed for long-range translocations. Both filaments are polarized, having a less dynamic minus end and a dynamic plus end, which is the primary site of growth and shrinkage. The minus ends of microtubules are attached to the microtubule-organizing center (MTOC), which is a highly organized macromolecular complex containing key proteins such as γ-tubulin, which interacts with tubulin monomers [86].

These cytoplasmic tracks are employed by two different transport principles: slow transport using the dynamics of filament polymerization and depolymerization from the plus to the minus end and rapid transport using motors. This type of transport is exemplified by chromosome movement during cell division and the retrograde transport of Baculoviruses to the nucleus [87].

Most translocations occur by molecular motors, which convert ATP hydrolysis into mechanical work. There are different classes of motors that use different filaments and execute movements in different directions. Myosins, encoded by 40 different genes in humans, are important for organelle movement, moving their cargo towards the plus end of actin filaments [88]. Kinesins represent a superfamily of 14 family members and are encoded by 40 different genes in humans. They are classically composed of a tandem of heavy chains, comprising the motor domain, which generates a force of 5 pN upon ATP hydrolysis [89]. The heavy chains are connected to a stretched stalk and a tail domain, with which cargo, including viruses, may interact for their anterograde transport (e.g., herpes simplex 1 capsids, vaccinia virus, pseudorabies virus, and SV40). However, kinesins are diverse, and light chains, adaptors, and other accessory molecules might be involved.

For viral infections in which a genome has to be transported to the cell nucleus (retrograde transport), dynein is of overwhelming importance. Dyneins (DYNs) are complexes of approximately 1.4 MDa and are composed of six different polypeptides (Figure 4A) [90]. Two heavy chains (HCs) of 520–530 kDa contain the motor domain, hydrolyzing ATP similar to the motor domain of kinesins. The HC tail binds to several types of associated subunits, including intermediate chains (ICs, 54, 59, and 74 kDa), light-intermediate chains (LICs), and three classes of light chains (LCs, around 20 kDa): T-complex testis-expressed protein (TCTEX), LC8, and Roadblock. All of the latter chains are involved in cargo binding and regulation of dynein function/activity.

#### 2.3.2. Retrograde Transport of Other Viruses

Dynein-mediated transport is not only crucial for various cellular processes but also in the transport of viral cargo. Viral examples include HSV-1 proteins pUL35 (VP26) and pUL46 (VP11/12), which recruit multiple active dynein motors to the HSV1 capsid by binding to DYN LC for retrograde transport towards the nucleus [91]. Of note, DYN binding does not necessarily result in continuous retrograde transport. Time-lapse microscopy of cytoplasmic HSV-1 capsids showed an overall movement towards the nucleus but with interruptions such as movement arrests and even short periods of anterograde transport, indicating a temporary switch of the molecular motor [92].

#### 2.3.3. HBV: Cytosolic Genome Release Versus Genome Transport in the Capsid

Considering that initial infection of hepatocytes is extremely efficient, a directed cytoplasmic transport must be hypothesized. The knowledge about the intracytoplasmic transport of HBV is, however, poor. Experimentally, cytoplasmic transport investigations are challenging, as probably only one virus enters the cell in vivo, while experimental cell-based systems require hundreds of virions. This causes difficulties when using imaging techniques, as a moving capsid is not necessarily linked to the infection process.

Evidently, intracytosolic HBV genome transport is intertwined with the site of genome release. If genome release occurs within the cytoplasm, the complex of pol with the covalently attached genome must be translocated to and into the nucleus, and, in fact, a nuclear localization signal (NLS) was identified on pol [93], which was exposed after phosphorylation by CK-2. CK-2 inhibition prevented not only the nuclear import of pol but also inhibited the establishment of infection, suggesting that the nuclear import of pol is important for the viral life cycle [93]. These data are in agreement with the findings of others, showing that the HBV DNA–pol complex enters the nucleus after its extraction from HBV virions by urea and its addition with cytoplasmic extract to isolated nuclei [94]. Cytoplasmic release of the HBV genome, which would be a pre-requisite for nuclear translocation without capsid involvement, is supported by the findings of others who observed protein-free HBV DNA in the cytoplasm [95,96]. This observation may indicate that the pol–genome complex is released and a part of the complexes is imported into the nucleus, while the pol of the other complexes is degraded. However, this scenario was not experimentally verified, and the study did not give clear evidence for the purity of the intracellular fractionated compartments.

Additional indirect support for cytoplasmic genome release comes from a study by Li et al. showing that HBV DNA can be sensed by a cytoplasmic Ku70/80 complex at least after transfection of HBV plasmid DNA, initiating cytokine CCL3 and CCL5 expression similar to that found in HBV-infected patients [97]. However, HBV is mostly considered to be a stealth virus, escaping significant detection by the innate immune response in chimpanzees and humans [98,99] and thus causing ambiguity about this pathway.

An alternative pathway is an active nuclear translocation of the HBV rcDNA within the viral capsid. Using three different kinds of capsids—capsids with rcDNA derived from cell culture HBV, RNA-containing capsids expressed in *E. coli*, and empty capsids—rcDNA-containing and empty capsids bound to in vitro polymerized microtubules indirectly in the presence of cytosolic proteins, which were identified as part of dynein [100]. Co-precipitation and co-sedimentation revealed the dynein light chain LL1 as a binding partner, and microinjection into the cytoplasmic periphery of *Xenopus leavis* oocytes showed active translocation to the nuclear pore complex (NPC). The active transport could be prevented by co-injection of a dynein LL1 mutant (DynLL1 H41Y), which binds to the capsids but not the dynein complex, indicating the need for capsid–dynein complex interaction for active transport.

What remains unclear—for HBV and for other viral cargo—is the fate after the dynein complex reaches its end at the MTOC. As a hypothesis, a switch of the capsids to kinesin is possible, given that capsids of different viruses accumulate not only at the nuclear envelope (NE) opposing the MTOC but also in distant areas. However, further research on this topic is required.

### 2.4. Nuclear Transport and Genome Release

#### 2.4.1. Principles of Active Nuclear Import

In non-dividing cells, the nucleoplasm is separated from the cytoplasm by the NE. Passing this barrier is essential for all cells, e.g., for importing transcription factors into the nucleus and for exporting mRNAs into the cytoplasm. Translocation requires passage through the NPCs, which are the only connection between the two compartments. In hepatocytes, the number of NPCs is estimated to be relatively high, ranging between 3 and 5000 NPCs/cell. There are few notable exceptions of viruses bypassing the NPCs: SV40 [101] and murine polyomavirus [102] appear to enter the ER from where they translocate to the inner nuclear membrane (INM), as the lumen of the ER is a continuum with the space between the inner and outer nuclear membrane (ONM). Porins created by the internal capsid proteins VP2 and/or VP3 may then lead to membrane degradation, allowing the exit of the genome either into the cytoplasm or into the nucleus.

Classical nuclear import is facilitated by nuclear transport receptors of the importin (also termed karyopherin) superfamily. This superfamily comprises a number of importins, including eight importin α molecules (importin α1–8); three transportins (transportin 1–3); importin 4, 5, and 7–11; and importin β1 (karyopherin β1) [103,104]. Importin α molecules are adaptor molecules, binding to their cargo via classical nuclear localization signals (cNLSs). This exposes the importin β1-binding domain (IBB), which is an approximately 40 amino acid-long arginine-rich domain, to importin β1, which then facilitates passage through the nuclear pore. The other importins and transportins interact with their cargo and facilitate nuclear import without adaptor molecules. However, non-karyophilic cargo may interact with a karyophilic protein, which then serves as an adaptor to nuclear transport receptors.

The interaction of the transport receptors with the cargo depends upon specialized signals exposed on the cargo’s surface. cNLSs comprise a mono- or bipartite stretch of basic amino acids (lysine (K), arginine (R)) and bind to importin α molecules [105]; PY-NLS and RGG domains bind to transportin 1 and 2 [106]; M9-NLS to transportin 1 [106]; and IPO4, 5, 7, and 13 recognition motifs interact with importin 4, 5, 7, and 13 [105]. Cargo exposing an IBB may directly interact with importin β1 [105]. Interaction with the nuclear import receptors may be modified by posttranslational modifications either within or adjacent to the NLS (e.g., phosphorylation inhibits interaction [107] or may result in conformational changes of the cargo, leading to exposure or hiding of the NLS, which can also be caused by interaction with other proteins.

The non-adaptor importins then interact with proteins of the nuclear pore, collectively called nucleoporins (Nups), in particular with those Nups containing phenylalanine-glycine (FG) repeats. The FG repeats form a hydrophobic mesh within the nuclear pore, normally preventing the diffusion of macromolecules [108,109,110]. The size of cargo able to pass the nuclear pore by diffusion is not clearly defined and depends upon the charge and shape of the molecule. The exact mechanism of how the cargo–importin complexes pass the nuclear pore remains unclear, and five different models have been proposed, all relying on interactions between the nuclear transport factors and Nup with FG repeats.

Irrespective of the respective transport model, the transport reaction is terminated in the nuclear basket, which is a filamentous structure on the nuclear side of the NPC [111]. Although not well analyzed for most of the import receptors, the nuclear import mediated via importin β1 (with and without importin α adaptors) was investigated in great detail [103]. Importin β1 in the complex with (importin α) and cargo binds to the protein Nup153, allowing an interaction between importin β1 and the protein Ran in its GTP-bound form. RanGTP dissociates the complex, allowing the cargo to diffuse deeper into the nucleus, while the importin β1–RanGTP complex is directly exported to the cytoplasm without diffusion into the nucleoplasm. Importin α, if part of the complex, is also dissociated from the cargo upon RanGTP interaction with importin β1, potentially involving nuclear proteins such as Npap60 or RBBP4 [112,113], which can promote the disassembly of the importin complex. Importin α is then recycled to the cytoplasm in a trimeric complex with RanGTP and the exportin Cas. The export of both importins, α and β1, is terminated by GTP hydrolysis by RanGAP1 (with the help of RanBP1 and/or Nup358 (also termed RanBP2), which are proteins on the cytoplasmic fibers of the NPC). RanGDP is then imported into the nucleus by binding to the protein NTF2, and this binding is dissociated by the exchange of GDP by GTP at the chromatin-bound protein RanGEF. The driving force for nuclear import (and export) is the gradient of RanGTP and RanGDP [114].

#### 2.4.2. Nuclear Import of Genomes of Other Viruses

This pathway—or parts of it—is highjacked by viruses for their genome import, sometimes without requiring nuclear transport receptors for their interaction with the NPC. While flexible complexes of genome and associated proteins may become directly imported using nuclear import factors—as is the case for the influenza virus segments, which use importin α/β1 binding to the viral proteins NP, PB1, PB2, and PA—viruses bearing their genomes inside a capsid are hindered by the sheer size of the capsid, which exceeds the maximal transport diameter of the nuclear pore, which is 39 nm [115].

HSV-1 capsids, having a diameter of 125 nm, acquire importin β1 during their trafficking to the NPC [116]. Importin β1 mediates attachment to the NPC in an oriented manner, followed by the capsid opening at the portal opposing the NPC, which requires ATP hydrolysis [117]. The genome inside the capsid is pressurized, leading to the injection of the viral DNA through the NPC, beginning with the DNA encoding the early genes [118].

In contrast, adenoviral capsids, having a diameter of around 90 nm, bind to the NPC due to direct interactions between the adenoviral hexon protein and a nucleoporin on the cytoplasmic fibers of the NPC [119,120] called CAN/Nup214, although interactions with other Nups that are heavily O-GlcNAcylated were shown to contribute. Capsid disassembly involves interaction with a number of proteins, such as the cellular export receptor CRM1, Nup358/RanBP2, and kinesin-1 [121]. Nup358/RanBP2 interacts with the viral protein IIIa [122], though it is not primarily needed for NPC attachment. Kinesin-1, which interacts with both the capsid and Nup358, contributes to the disassembly process, probably by causing a mechanical force that tears the capsid apart [121]. This results in the release of the adenoviral DNA in complex with several viral proteins. This includes the covalently genome-bound terminal protein, protein IVa2, protein X (Mu), protein V, and protein VII [123,124]. The latter three are highly basic proteins rich in arginine. While protein V dissociates from the viral genome without entering the nucleus with the genome, protein VII mediates the nuclear import of the genome [124]. The required nuclear transport receptors are not fully identified, but the nuclear translocation of protein VII upon its expression in cell lines may occur by using importin α, importin β1, importin 7, and transportin [121,124].

There are just three animal viruses with capsids below the transport limit of the nuclear pore, including HBV, parvoviruses (18–28 nm), and circoviruses infecting pigs (15–22 nm). While it is known that the circoviral capsid protein of all porcine circoviruses, Cap, comprises cNLS [125], practically nothing is known about the nuclear import of the genome. Parvoviruses are better investigated, showing cNLS in the N-terminus of the capsid protein VP1, which becomes exposed upon acidification during clathrin-mediated endocytosis [126,127]. The acquisition of importin β1 occurs during cytoplasmic shuttling towards the nucleus [128], and importin binding was found to be essential for infection. Despite these findings supporting the passage of the capsids through the NPC-like cellular karyophilic cargo, parvoviral capsids also exhibit a direct Nup-binding capacity, as shown for H-1PV and adeno-associated virus 2 (AAV2) [129]. In contrast with what is known for other viruses, binding to the NPC triggers temporary holes in the NE, leading to exposing high Ca^++^ concentrations to the capsid. Capsid disassembly and genome release appear in a two-step mechanism consisting of an initial opening of the capsid upon Ca^++^ exposure, which is followed by diffusion of the capsid deeper into the nucleus. Genome release—at least of AAV2 vectors—is executed by interaction with proteins of the DNA damage response, namely by Rad52 decamerization, which physiologically occurs in the vicinity of the host’s chromatin.

A further exception is baculoviruses, which have rod-shaped capsids with a size of approximately 40 × 250–300 nm and which also deliver their genome within the capsid through the NPC, requiring a positioning at the NPC [130,131].

#### 2.4.3. Nuclear Import of HBV Genomes

The four arginine clusters in the CTD do not represent the typical cNLS, which is composed of similarly charged lysine residues. Nonetheless, they act as three bi-partite, overlapping NLS, which was shown by fusing them to BSA or by their potential to compete for the nuclear import of other cNLS-bearing cargo (aa142–155, aa158–168, aa165–175) [41,49,50,132,133]. Phosphorylation between the arginine clusters diminishes the importin α interaction, at least of the overlapping cNLS. Further, the exposure of the entire CTD enables direct importin β1 binding, replacing importin α binding due to the higher affinity of importin β1. Similar to importin α binding, the interaction of the CTD with importin β1 is reduced by CTD phosphorylation, as shown by using *E. coli*-expressed, in vitro-reassembled capsids that are SPRK-1-phosphorylated [74].

Binding to the nuclear transport receptors mediates binding to the NPC, as shown by using digitonin-permeabilized cells to which various capsids, including rcDNA-containing capsids, were added, together with importin α/β1 [49]. The system further allows the discrimination of capsids from non-assembled Cp, which is difficult in infected or transfected cells. In vitro-phosphorylated, RNA-containing capsids, which expose phosphorylated CTDs, also interacted with NPC via importin α/β1 [49], leading to the hypothesis that phosphorylation either results in the exposure of the neighboring non-phosphorylated CTD or that a cNLS neighboring the phosphorylated site acts as a ligand for importin α. For non-assembled subunits, these findings suggest that they can be imported directly via importin β1.

Nuclear import, including the nuclear transport receptors, is evolutionarily well conserved, which allows the use of even distant systems such as *Xenopus laevis* oocytes [131]. As pointed out for the cytoplasmic transport of HBV capsids, this system has the advantage of the sheer size of the cells, allowing for easier manipulation and the application of higher volumes. The microinjection of HBV capsids into *Xenopus laevis* oocytes was thus also used to investigate events at the NPC by electron microscopy. The results showed that capsids exposing the CTD (in vitro-phosphorylated capsids, rcDNA-containing capsids, and capsids containing replication intermediates but not rcDNA) not only bound to the cytoplasmic face of the NPC but were also observed within the nuclear basket [131]. As confirmed by biochemical assays, this binding was caused by interactions with the C-terminal domain of Nup153 [134].

Analyzing the fate of the different capsid species by confocal laser scanning microscopy showed, however, surprising results. As shown by using digitonin-permeabilized cells, in vitro-phosphorylated capsids expressed in *E. coli* co-localized with the NPCs without entering the nucleoplasm [49]. The same phenotype was observed using capsids from transfected cells, which contained replication intermediates but not rcDNA. These findings were in agreement with the capsid interaction with Nup153, which was much stronger than for importin β1, explaining why there was no displacement by physiological cargo [134].

In contrast, the addition of rcDNA-containing capsids resulted in nuclear capsid stain, which was combined with the appearance of released viral genomes as documented by fluorescence in situ hybridization under native conditions [135]. Cross-linking rcDNA-containing capsids resulted in co-localization with the NPC and entry into the nuclear basket (by electron microscopy) but a failure of the capsids to enter the karyoplasm. These findings allowed the hypothesis that the rcDNA-containing capsids had to fall apart in the basket to allow the entry of Cp subunits into the nucleoplasm. This interpretation was further supported by adding rcDNA-containing capsids, together with importin α and importin β1 but without RanGTP, to the assay, which also showed intranuclear capsid stain and intranuclear released rcDNA [134]. This indicated that the nuclear capsid stain was derived from Cp subunits, which had not to be dissociated from Nup153 by dissociating their indirect binding to Nup153 via importins by RanGTP.

Using anti-Cp/capsid antibodies with different assembly specificities showed that the intranuclear capsid stain was derived from intact capsids that re-assembled in the nucleus from Cp dimers, as shown by adding wheat germ agglutinin, which is a broad-range inhibitor of nuclear import and export [72]. The finding was confirmed by Nycodenz gradient centrifugation of the intranuclear capsids, further showing that these capsids contained RNA instead of rcDNA. Using the same experimental set-up but digesting the intranuclear RNA revealed that nuclear assembly was inhibited. Of note, these experiments showed that 99% of capsids added to the digitonin-permeabilized cells released their rcDNA, arguing that nuclear import and rcDNA release—at least in digitonin-permeabilized cells—was highly efficient.

Collectively, these findings suggest that CTD-exposing capsids enter the nuclear basket via the importin pathway but get stuck in the nuclear basket by their interaction with Nup153. rcDNA-containing capsids then disintegrate into Cp subunits and release the viral genome by molecular mechanisms that have yet to be elucidated but may involve Cp sumoylation [81]. Cp subunits, which are not in direct contact with Nup153 and exist in only 16 copies in the basket, then diffuse deeper into the karyoplasm, where they re-assemble into capsids containing cellular RNAs. Of note, the vast majority of nuclear capsids are not derived from genome transport but are assembled from excessively expressed Cp, which are phosphorylated. This notation is in agreement with the low number of HBV genomes per infected hepatocyte and the observation that nuclear capsids are devoid of nucleic acids. However, in light of the findings of deproteinized cytoplasmic rcDNA, a partial capsid disassembly in the cytoplasm with the subsequent nuclear import of another non-capsid rcDNA–Cp complex cannot be excluded. As shown by its resistance against the cytoplasmic exonuclease TREX1, the rcDNA in such a complex must, however, still be protected [136].

### 2.5. Repair of the HBV Genome

The finding that rcDNA-containing capsids release the viral genome but capsids with replication intermediates do not has not yet been explained. The differences may comprise their different phosphorylation levels, as phosphorylation modulates electrostatic interactions within the capsid. It was described that dephosphorylation can prompt capsid disassembly [80]; however, this effect might also be caused by the nature of the encapsidated nucleic acid, as capsids with mature rcDNA exhibit a low phosphorylation pattern. Lower stability may, in turn, lead to exposure of parts of the genome, resulting in the hypothesis that repair protein(s) might participate in genome release.

The structure of the rcDNA but also dslDNA requires a number of distinct repair processes (Figure 5A). This includes the removal of the polymerase, capped RNA primer, and terminal redundancies; completion of the plus-strand DNA; and ligations of the 5′ and 3′ ends of minus- and plus-strand DNA. Detailed analyses using various approaches showed that distinct sets of enzymes were required to repair minus- and plus-strand DNA and that they differ between rcDNA and dslDNA repair.

Biochemical studies using recombinant rcDNA revealed that Flap endonuclease 1 (FEN-1) for removing pol and terminal redundancy and DNA ligase-1 (LIG-1) are sufficient to repair minus-strand DNA, while plus-strand DNA repair needs proliferating cell nuclear antigen (PCNA), replication factor C (RFC) complex, and DNA polymerase δ (POLδ), aside from FEN-1 and LIG-1 [137]. RFC complex recognizes the 3′ end of the plus-strand DNA and loads PCNA onto the prime-template junction of the plus strand [138]. PCNA then recruits and activates POLδ [139] and replaces the RNA primer, which is executed by FEN-1 [140]. However, some of the enzymes might be complemented or replaced by other host cell factors. This is indicated by loss-of-function experiments, which showed the participation of pol κ, η, and plus-strand DNA completion, although cccDNA formation experiments using yeast extracts revealed that POLδ could not be replaced by other cellular polymerases. In this system, FEN-1 was also not required [141].

The chronology of these events is not fully understood, but FEN-1 action requires the creation of a flap, which is thought to be caused by POLδ replacing the RNA primer from the minus-DNA strand [137], which, in turn, requires RFC complex and PCNA.

However, other host proteins, such as Tyrosyl-DNA phosphodiesterase 2 (TDP2), were also shown to remove the viral polymerase from the 5′ end of the minus-strand DNA, although its knock-out still allowed some cccDNA formation, probably by other cellular proteases. Pol removal could be then followed by removal of the terminal redundancy promoted by the ataxia-telangiectasia-mutated-and-Rad3-related kinase (ATR) and its major downstream effector checkpoint kinase 1 (CHK1) [142].

Further, functionally less characterized proteins, such as DDB1 (DNA Damage Binding protein 1), identified by mass spectrometry and ChIP-qPCR [143]; and Y box binding protein 1 (YBX-1) have been described. Of note, YBX-1 expression correlates with both HBV load and liver disease progression, further supporting its significance in the HBV life cycle [144].

Based on in cellulo experiments using DHBV, another pathway of cccDNA formation from dslDNA was proposed (Figure 5B) [145]. Here, KU-80, a component of the non-homologous end joining (NHEJ) repair pathway, and ligase-4 (LIG-4) were found to be necessary for cccDNA formation based on deproteinized dslDNA. It remains, however, open as to whether the circularization results in functional cccDNA, which would require the precise removal of the redundancy, or to what extent this pathway contributes to cccDNA formation.

## 3. Summary

The fate of the hepadnaviral genome is closely linked to the structural dynamics of its capsid, which undergoes changes based on the encapsidated nucleic acid and the phosphorylation state of the capsid. Capsid trafficking, while not entirely understood, is influenced by the multifunctional CTD of the Cp. This domain mediates interactions with DYN LL1 for retrograde transport within the cytoplasm, as well as nuclear import receptors for genome delivery to the nucleus.

Hepadnaviruses exhibit a unique genome trafficking strategy among animal viruses. The capsid-encased viral genome can be transported into the nucleus for transcription or enveloped for progeny virus assembly, highlighting the lack of distinct pathways for infection and virus production. Notably, the ability of the genome to cross the nuclear envelope within intact capsids is an uncommon feature shared by only a few viruses. Evidence also suggests the potential for capsid disassembly and genome release at the nuclear basket, a process that appears unique and remains uncharacterized in other viruses.

### Therapeutic Implications

Most cellular interaction partners involved in HBV capsid trafficking—such as importins, dyneins, protein kinases, and nuclear pore proteins—are essential for cell viability, making them challenging therapeutic targets due to the high likelihood of significant side effects, as evidenced by anti-cancer drugs that target these proteins. A notable exception is ivermectin, which targets the importin α/β1 pathway and is used to treat certain parasitic infections, and it exhibits limited side effects. Nevertheless, given the low toxicity of current therapies, such approaches may not be practical. This highlights the potential of targeting the HBV core protein (Cp), particularly its C-terminal domain (CTD), as a promising druggable target due to its critical role in multiple essential processes during HBV infection. Moreover, drugs that induce cytoplasmic Cp or capsid accumulation could facilitate Cp degradation, potentially enhancing the presentation of T-cell epitopes on hepatocyte surfaces and promoting the immune-mediated elimination of HBV-infected cells.

## Figures and Tables

**Figure 1 microorganisms-13-00157-f001:**
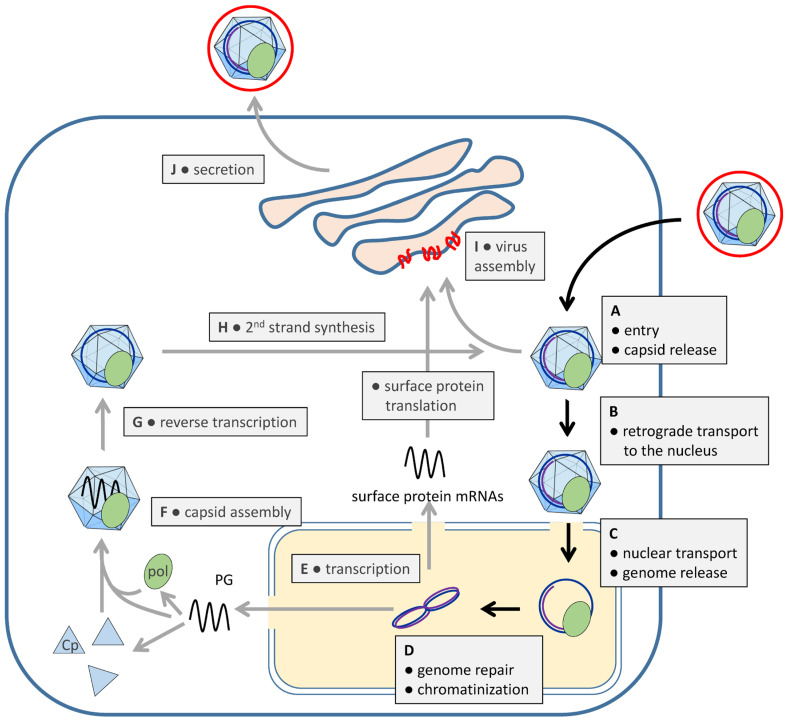
Schematic overview of the HBV life cycle. The cell membrane is shown as a bold blue line; the nuclear envelope is indicated by a double blue line. Cytoplasm: white; nucleoplasm: yellow; ER/Golgi: light red. HBV surface proteins: red; viral polymerase: green. The different steps are indicated in the figure. For further details, see text.

**Figure 2 microorganisms-13-00157-f002:**
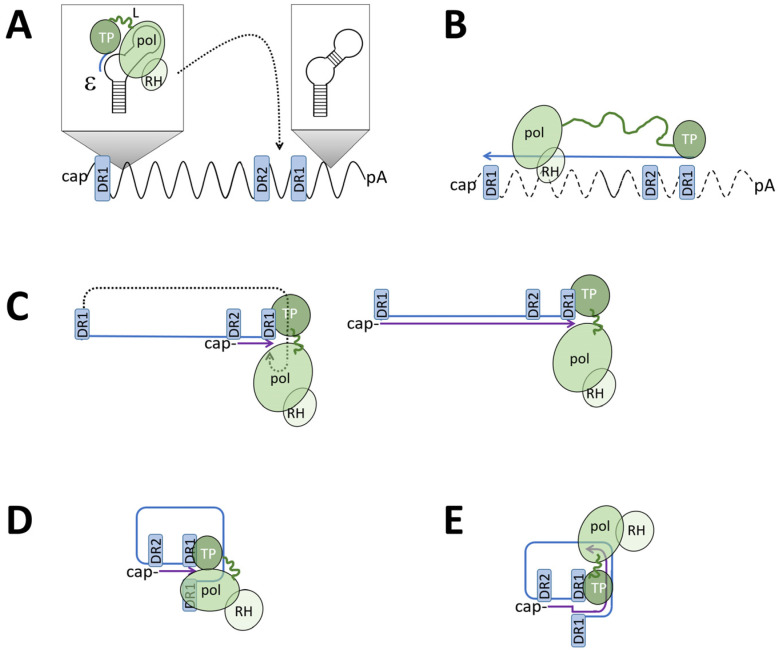
HBV genome maturation. PG: black wave line; minus-strand DNA: blue line; plus-strand DNA: violet line. TP: terminal protein (=pol priming domain); pol: catalytic pol domain; RH: pol RNase H domain. The linker between TP and the catalytic domain is shown as a green line of variable length. DR1: direct repeat 1 present in 2 copies on the PG. DR2: direct repeat 2. (**A**) Pol binds to e and synthesizes the first nucleotides, causing a covalent link to the TP of pol. This complex translocates to DR1. (**B**) Pol synthesizes the minus-strand DNA until the 5′ end of the PG, which has the DR1 sequence. Synthesis is combined with degrading the RNA from the RNA–DNA hybrid. The first 11–16 nucleotides of the RP (including cap) stay non-degraded. (**C**) **left**: The RNA fragment translocates to DR2 upstream of DR1, where pol initiates plus-strand DNA synthesis. **Right**: the RNA primer translocation does not occur (approx. in 10%) and pol synthesizes the plus-strand DNA, resulting in dslDNA. (**D**) When reaching the end of the minus-strand DNA, the 5′ end of the plus strand anneals to the complementary DR1 on the 5′ end of the minus-strand DNA. (**E**) Pol continues with plus-strand DNA synthesis, creating a circular form of the genomes (rcDNA). The plus-strand DNA stays incomplete.

**Figure 3 microorganisms-13-00157-f003:**
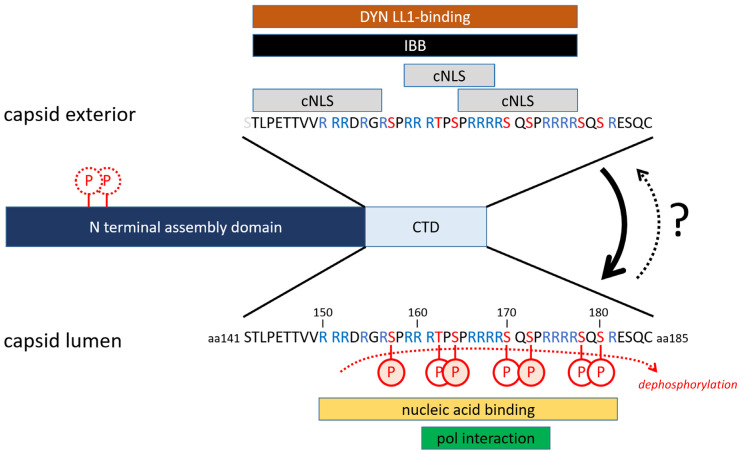
Domains of the core protein. Assembly domain: light green; CTD: orange. Red P: phosphorylation sites (only shown for internal CTD localization). Orange-filled P: major phosphorylation sites. The figure shows the interaction partners and functions of the CTD for its internal and external location in the assembled capsid. For the effect of phosphorylation, see text.

**Figure 4 microorganisms-13-00157-f004:**
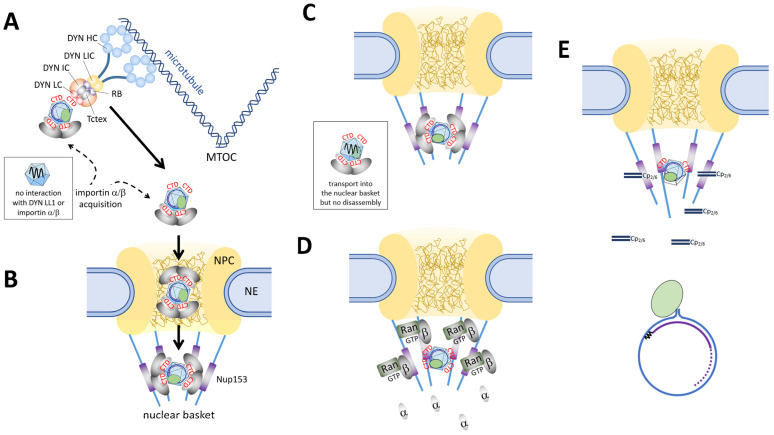
Schematic overview of cellular capsid transport. The boxes indicate capsids having other properties. (**A**) Cytoplasmic transport. CTD exposure is shown as red CTD, and the dynein chains are indicated. Importins are shown as grey ovals. (**B**) Passage through the nuclear pore, filled with FG repeats (brown lines). (**C**) Arrest in the nuclear basket by interaction with Nup153 (purple). (**D**) Dissociation of importin α/β from the capsid and capsid binding to Nup153. (**E**) Capsid disassembly and genome release. Released Cps are depicted as dimers or hexamers. Released rcDNA: blue line; minus-strand DNA: black wavy line; RNA primer for plus-strand DNA synthesis: violet line; plus-strand DNA with a variable 3′ end (dotted line). Pol: green oval.

**Figure 5 microorganisms-13-00157-f005:**
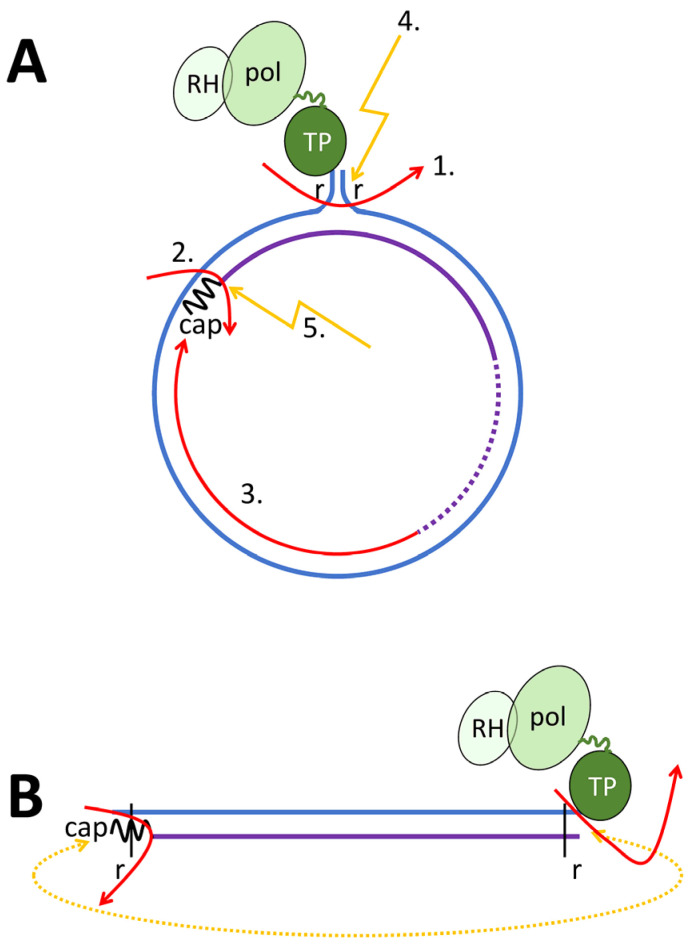
Schematic structure of the rcDNA and the repair processes leading to cccDNA. Minus-strand DNA: blue line; “r”: terminal redundancy of the minus-strand DNA; RNA primer for plus-strand DNA synthesis: black wave line; plus-strand DNA: violet line with a variable 3′ end indicated by a dotted line. TP: terminal protein (=pol priming domain); pol: catalytic pol domain; RH: pol RNase H domain. The linker between TP and the catalytic domain is shown as a green line of variable length. (**A**) rcDNA. 1. Removal of the minus-strand DNA redundancy “r”. 2. Removal of the RNA primer. 3. Completion of the plus-strand DNA. 4. Ligation of 5′ and 3′ ends of the minus-strand DNA. 5. Ligation of 5′ and 3′ ends of the plus-strand DNA. (**B**) dslDNA. Removal of pol, RNA primer as in A. Circularization of the molecule, mediated by KU70, is indicated by a dotted orange line.

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
