# Peer review of "From the Cytoplasm into the Nucleus—Hepatitis B Virus Travel and Genome Repair"

_microorganisms, 2025, doi:10.3390/microorganisms13010157_

Round 1

Reviewer 1 Report

Comments and Suggestions for Authors

In this review, Ringlander et al. systematically explored the current knowledge of HBV intracytoplasmic and nuclear transport, as well as genome repair processes. The topics are highly attractive as these steps of the HBV life cycle are complex and currently mysterious. I have some issues outlined below.

1.Clarification on Core Protein and cccDNA: Lines 95-96 state, “Although the details are not fully understood, the core protein (Cp) may also interact with cccDNA, influencing transcription.” However, more evidences (PMID: 36226986, PMID: 36840581) suggest that HBV core protein has no effect on cccDNA transcription. This section should be updated to reflect these findings more comprehensively.

2.Text Formatting Issues: There are numerous abnormal symbols (resembling a vortex) in the text, such as on lines 105 and 106. These should be corrected for clarity and professionalism.

3.Simplification of Technical Sections: Some sections, like those on nuclear transport mechanisms, are highly detailed and could benefit from summarization to maintain readability for a broader audience.

4.Therapeutic Implications: Discuss the potential translational impact of understanding HBV capsid dynamics and genome repair, especially regarding therapeutic targeting of cccDNA.

Author Response

Reviewer #1

In this review, Ringlander et al. systematically explored the current knowledge of HBV intracytoplasmic and nuclear transport, as well as genome repair processes. The topics are highly attractive as these steps of the HBV life cycle are complex and currently mysterious. I have some issues outlined below.

1.Clarification on Core Protein and cccDNA: Lines 95-96 state, “Although the details are not fully understood, the core protein (Cp) may also interact with cccDNA, influencing transcription.” However, more evidences (PMID: 36226986, PMID: 36840581) suggest that HBV core protein has no effect on cccDNA transcription. This section should be updated to reflect these findings more comprehensively.

Response: We thank the reviewer for the additional citation, which were added to the text (lines 101/102).

2.Text Formatting Issues: There are numerous abnormal symbols (resembling a vortex) in the text, such as on lines 105 and 106. These should be corrected for clarity and professionalism.

Response: We excuse for these formatting problems, which apparently occurred during the transfer from the original word file to the “microorganisms” template.

3.Simplification of Technical Sections: Some sections, like those on nuclear transport mechanisms, are highly detailed and could benefit from summarization to maintain readability for a broader audience.

Response: We accept that the paragraph includes many details, which are possibly not directly linked to HBV nuclear import. However, many details refer to specialized proteins involved transport receptor shuttling such as RanGTP (described in lines 439 – 451. Their function is essential for understanding key experiments for HBV nuclear import e.g. that the nuclear import of HBV genomes is independent upon RanGTP, which strongly supports that capsid disassembly occurs within the nuclear basket (lines 548 – 553). The same is true for the interaction of importin beta with importin beta-binding domains as exposure of the full CTD thus of an IBB may indicate that empty capsids show different nuclear pore interactions than capsids derived from the virion.

Other transport receptors were mentioned as they are functionally redundant, and that an impact of them on nuclear capsid interaction with the NPC is not studied yet.

Other, possibly too detailed information, was added for describing the knowledge on other viruses. The detailed descriptions were added to give the reader an idea of that what is known about other viral examples and to allow the comparison between transport strategies including the complexity of the topic. However, details about the interactions of HSV and adenoviruses were removed from the text, as long as they do not demonstrate direct parallels or differences to HBV (formerly lines 468 – 475 and lines 479 – 484).

4.Therapeutic Implications: Discuss the potential translational impact of understanding HBV capsid dynamics and genome repair, especially regarding therapeutic targeting of cccDNA.

Response: We added a short section on therapeutic implications at the end of the manuscript.

Reviewer 2 Report

Comments and Suggestions for Authors

The manuscript by Ringlander et al. provides a comprehensive review of published studies concerning the mechanisms of intracellular transport and genome repair in Hepatitis B Virus. Hepatitis B is a significant global health concern, making HBV research a priority in virology. The authors have conducted a thorough analysis of approximately 150 articles, presenting a well-illustrated and structured review. The manuscript is well-written and engaging, prompting consideration of the evolutionary basis for the unique aspects of HBV trafficking mechanisms and their relationship to the HBV capsid protein structure. This review has the potential to be of broad interest to virologists, not only those specializing in the field. Therefore, I believe the manuscript is suitable for publication in Microorganisms. However, addressing the following points would further enhance its quality:

Literature Review: The reviewed literature could benefit from the inclusion of more recent articles.

Terminal Protein: While the replication processes are well-described, a discussion of the role of the terminal protein is lacking.

Recent Cryo-EM Study: The manuscript should incorporate and discuss the recent, compelling cryo-EM study of hepatitis B and woodchuck hepatitis virus by Liu et al. (2022) (https://www.science.org/doi/10.1126/sciadv.abo4184).

Line 74: Please provide an appropriate reference to support the statement regarding HBV entry into hepatocytes via clathrin-dependent endocytosis following interactions.

Formatting: The manuscript requires a thorough review to correct formatting inconsistencies, including missing spaces before references.

Author Response

Reviewer #2

The manuscript by Ringlander et al. provides a comprehensive review of published studies concerning the mechanisms of intracellular transport and genome repair in Hepatitis B Virus. Hepatitis B is a significant global health concern, making HBV research a priority in virology. The authors have conducted a thorough analysis of approximately 150 articles, presenting a well-illustrated and structured review. The manuscript is well-written and engaging, prompting consideration of the evolutionary basis for the unique aspects of HBV trafficking mechanisms and their relationship to the HBV capsid protein structure. This review has the potential to be of broad interest to virologists, not only those specializing in the field. Therefore, I believe the manuscript is suitable for publication in Microorganisms. However, addressing the following points would further enhance its quality:

Literature Review: The reviewed literature could benefit from the inclusion of more recent articles.

Response: 5 newer citations have been added.

Terminal Protein: While the replication processes are well-described, a discussion of the role of the terminal protein is lacking.

Response: We added more detailed information about the function of the terminal protein within the replication cycles in lines 114 – 124.

Recent Cryo-EM Study: The manuscript should incorporate and discuss the recent, compelling cryo-EM study of hepatitis B and woodchuck hepatitis virus by Liu et al. (2022) (https://www.science.org/doi/10.1126/sciadv.abo4184).

Response: We thank the reviewer for this comment. We added the information in this publication in lines 60 - 63.

Line 74: Please provide an appropriate reference to support the statement regarding HBV entry into hepatocytes via clathrin-dependent endocytosis following interactions.

Considering that HBV entry was not the central part of the review, the authors decided to not include the citation and its discussion.

Formatting: The manuscript requires a thorough review to correct formatting inconsistencies, including missing spaces before references.

Response: We excuse for these formatting problems, which apparently occurred during the transfer from the original word file to the “microorganisms” template.

Reviewer 3 Report

Comments and Suggestions for Authors

The authors in this manuscript have reviewed the life cycle of Hepatitis B virus (HBV) by focusing on the principles of nuclear transport of viral cargos and viral genome repair. The review was well-prepared by summarizing the current findings in detail, and with proper revision, the manuscript would be a good candidate for the publication in the journal. Please revise the manuscript thoroughly and carefully since some flaws significantly affect the overall paper quality.

1 Multiple typos, grammatical and format errors are seen throughout the manuscript. For example, a wavy underline is seen in Figure 1, various special symbols are not shown correctly, some brackets only have the left part, etc.

2 The format of Figure 4 needs to be updated since all panel are too big to be arranged neatly.

3 The section of 2.4.2 is too redundant which concludes the nuclear transport of other viruses. Please make this section concise and highlight the section of 2.4.3 (for HBV).

4 Can you add your own perspectives in the final Summary section, especially the future research directions and possible clinical applications such as drug design?

Author Response

Reviewer #3

The authors in this manuscript have reviewed the life cycle of Hepatitis B virus (HBV) by focusing on the principles of nuclear transport of viral cargos and viral genome repair. The review was well-prepared by summarizing the current findings in detail, and with proper revision, the manuscript would be a good candidate for the publication in the journal. Please revise the manuscript thoroughly and carefully since some flaws significantly affect the overall paper quality.

1 Multiple typos, grammatical and format errors are seen throughout the manuscript. For example, a wavy underline is seen in Figure 1, various special symbols are not shown correctly, some brackets only have the left part, etc.

Response: We excuse for these formatting problems, which apparently occurred during the transfer from the original word file to the “microorganisms” template.

2 The format of Figure 4 needs to be updated since all panel are too big to be arranged neatly.

Response: The figure was re-arranged and placed later in the text (now line 579).

3 The section of 2.4.2 is too redundant which concludes the nuclear transport of other viruses. Please make this section concise and highlight the section of 2.4.3 (for HBV).

Response: Details about the interactions of HSV and adenoviruses were removed from the text, as long as they do not demonstrate direct parallels or differences to HBV (formerly lines 468 – 475 and lines 479 – 484).

4 Can you add your own perspectives in the final Summary section, especially the future research directions and possible clinical applications such as drug design?

Response: We added a section on therapeutic implications at the end of the manuscript.

Round 2

Reviewer 3 Report

Comments and Suggestions for Authors

The authors have addressed all my concerns and revised most of the flaws. Now the overall quality of manuscript has been improved which would be suitable for acceptance and publication.